# EXECUTABLE COUNTERFACTUALS: IMPROVING LLMs' CAUSAL REASONING THROUGH CODE

**Aniket Vashishtha** [1*]   **Qirun Dai** [2*]   **Hongyuan Mei** [3]
**Amit Sharma** [4†]   **Chenhao Tan** [2†]   **Hao Peng** [1†]
[1]University of Illinois Urbana-Champaign    [2]The University of Chicago
[3]TTIC    [4]Microsoft Research India

## ABSTRACT

Counterfactual reasoning, a hallmark of intelligence, consists of three steps: inferring latent variables from observations (*abduction*), constructing alternative situations (*intervention*), and predicting the outcomes of the alternatives (*prediction*). This skill is essential for advancing LLMs' causal understanding and expanding their applications in high-stakes domains such as scientific research and healthcare. However, existing efforts in assessing LLM's counterfactual reasoning capabilities tend to skip the abduction step, effectively reducing to interventional reasoning and leading to over-estimated LLM performance. To address this, we introduce *executable counterfactuals*, a novel framework that operationalizes causal reasoning through code and math problems. Our framework explicitly requires all three steps of counterfactual reasoning and enables scalable synthetic data creation with varying difficulty, creating a new frontier for evaluating and improving LLM's reasoning. Our results reveal substantial drop in accuracy (25-40%) from interventional to counterfactual reasoning for state-of-the-art models such as *o4-mini* and *Claude-4-Sonnet*. To address this gap, we construct a training set comprising counterfactual code problems having *if-condition* and test on out-of-distribution code structures (e.g., having *while-loop*); we also test whether a model trained on code can generalize to counterfactual math word problems. While supervised finetuning (SFT) on stronger models' reasoning traces improves in-distribution performance of Qwen models, it leads to a *decrease* in accuracy on out-of-distribution tasks. In contrast, reinforcement learning (RL) induces the core cognitive behaviors and generalizes to new distributions, yielding substantial accuracy gains over the base model on both code (↑~1.5–2X) and counterfactual math problems. Analysis of the reasoning traces further reinforces these findings and highlights the promise of RL with scalable data generation for improving LLMs' counterfactual reasoning. Our code and data are available at `https://github.com/AniketVashishtha/Executable_Counterfactuals`.

## 1 INTRODUCTION

Counterfactual reasoning is the cognitive process of answering *what-if* questions that underpin critical domains such as scientific discovery (Schölkopf et al., 2021), healthcare (Richens et al., 2020), economics (Athey & Imbens, 2017), and public policy (Poulos & Zeng, 2021). Given an action and an observed outcome, it involves inferring the latent state of a system when the action was performed (*abduction*), constructing alternative scenarios through *interventions*, and *predicting* the outcomes under those counterfactual scenarios (Pearl, 2002a; Epstude & Roese, 2008). Despite the importance of counterfactual reasoning, it remains a widely documented weakness of current large language models (LLMs; Jin et al., 2024; Yamin et al., 2025; Yu et al., 2023).

Evaluating and improving counterfactual reasoning is challenging because counterfactuals are inherently unobservable and rely on hypothetical alternatives to reality. As a result, prior work considers

---

[*]Equal contribution.

[†]Equal advising.
 Correspondence to: `aniketv2@illinois.edu, qirundai@uchicago.edu`.

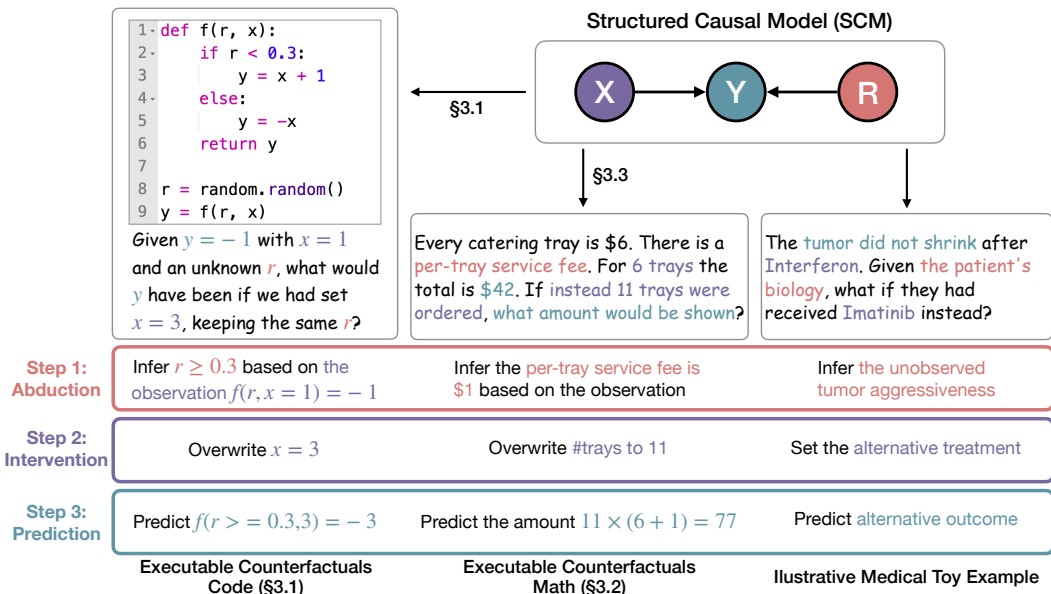

Figure 1: Our Executable Counterfactuals framework is rooted in code reasoning (left), and can also be easily extended to the math domain (middle). Code offers a controlled, executable setting that maps naturally to causal/computational graphs and transfers to natural-language tasks. We also offer a medical toy example (right) to better illustrate the three cognitive skills, abduction–intervention–prediction, required for addressing true counterfactual reasoning.

either synthetic graph-based settings that are hard to map to real-world problem solving (Jin et al., 2024) or simplistic tasks that are *expressed* in counterfactual language but can be solved without invoking all aspects of counterfactual reasoning. Examples include binary classification tasks given full information about the causal graph (*Would $Y$ still occur if $X$ didn't happen?*; Chen et al., 2025) or benchmarks based on perturbations of existing reasoning problems (thus creating new "counterfactual" problems; Wu et al., 2024). With full information (i.e., there are no latent confounders or noise), these problems can be solved by simple forward reasoning (Gerstenberg, 2022): change the input variables' values as instructed and solve it as a new problem, *without* any counterfactual reasoning.

These simplified interpretations of counterfactuals risk conflating them with simpler forms of causal reasoning (§2) and thus misrepresent LLMs' counterfactual abilities. To address these limitations, we identify the three core cognitive skills from Pearl's definition of counterfactual reasoning (Pearl, 2002a)—*Abduction* , *Intervention* and *Prediction*—and construct tasks that requires all three skills to obtain a correct solution. Key benefits of this perspective include explicit separation of counterfactual reasoning from simpler forms of causal reasoning, fine-grained attribution of models' strengths and weaknesses, and an actionable framework for improvement (§3). Moreover, beyond counterfactuals, improvements to these cognitive skills can independently serve as building blocks for stronger LLM reasoning in general.

Our key idea is to use code understanding as the problem setup to study counterfactual reasoning (i.e., *executable counterfactuals*). Specifically, we investigate whether a model can correctly reason about the output of a code function when its input arguments are differently set, conditioned on a pair of factual input and output. Notably, we introduce latent random variables to these code functions. The values of these variables are kept unchanged throughout the reasoning process, but not revealed to the language model, thus requiring the model to infer them from the given factual inputs and outputs. In the framework of structural causal models, these latent variables can be considered as noise variables that need to be inferred from the given factual information before making any counterfactual prediction. As shown in the illustrative example in Figure 1, the causal structure $X \to Y \leftarrow R$ where $X$ and $R$ independently cause $Y$ converts to a program where $X$ computes $Y$ while $R$ determines conditional branching. A counterfactual question is constructed as: *Given observation $y = f(r, x = 1) = -1$ with unknown $r$, what would $y$ have been if we had set $x = 3$, keeping the same $r$?* Solving this problem requries the agent to invoke all three cognitive skills: (1)

infer $r$ based on the observation $x = 1, y = -1$ (abduction), (2) mentally set $x = 3$ (intervention), and (3) compute the resulting $y$ (prediction).

Beyond the aforementioned benefits, our code-based framework avoids the potential ambiguity of natural language, and allows rich and controllable complexity for constructing evaluation problems and generating synthetic training data. It evaluates models' ability to use counterfactual reasoning for problem solving rather than reducing the task to answer binary classification questions (Jin et al., 2024; Chen et al., 2025). In addition, it facilitates evaluating and improving out-of-distribution generalization by varying the program structures and translating coding tasks into math problems (§3.2). We address the following important research questions with executable counterfactuals:

1. ***How do current LLMs perform on counterfactual reasoning?*** Our experiments with open models of sizes ranging from 1.5B to 72B parameter and commercial reasoning models show strong performance on straightforward code-execution tasks, but poor performance on counterfactual reasoning over the same code. Qualitative analysis indicates consistent failure at the abduction step, leading to incorrect conditioning on the original observation to infer latent features.

2. ***Can SFT distillation from stronger models instill these skills and do they generalize?*** We finetune Qwen 1.5B/3B/7B-Instruct on reasoning trajectories from DeepSeek-Distilled-Qwen-32B, and observe ~40% performance improvements on in-domain evaluation. However, these improvements do not generalize to unseen code structures or counterfactual math problems, highlighting the limited generalization of SFT.

3. ***How does RL fare?*** Training the same models with RL from verifiable rewards (RLVR) using GRPO (Shao et al., 2024) leads the models to acquire the necessary cognitive skills, showing strong transfer across diverse code structures and counterfactual math problems in natural language, with concrete evidence of improved generalization.

Our findings have two key implications. First, they reinforce recent evidence that current LLMs remain weak at counterfactual and causal reasoning (Jin et al., 2024; 2023; Willig et al., 2023). Second, our experiments call into question the effectiveness of SFT, a widely adopted approach by recent works to improve counterfactual reasoning (Guo et al., 2025; Li et al., 2025), especially regarding its ability to generalize to complex and high-impact real-world domains. In contrast, our results show that RL elicits stronger generalization for counterfactual reasoning; despite training only on code, the model internalizes the core skills and applies them directly to counterfactual math problems, providing early evidence that RL is a promising pathway for eliciting such reasoning in LLMs. Crucially, as shown in the experiments, our code-based framework has the potential to offer a scalable way for learning counterfactual reasoning that transfers to new domains where training data can be scarce. All code and data will be publicly released upon publication.

## 2 BACKGROUND AND RELATED WORK

In this section, we will first outline the cognitive skills required for counterfactual reasoning and then show how it is often conflated with interventional reasoning in prior work.

**From abduction to prediction.** We use Figure 1 as a running example to expand the cognitive skills required for counterfactual reasoning. Three steps are needed to answer the counterfactual question *Given observation $y = f(r, x = 1)$, what would $y$ have been had $x = 3$ instead in the original run?*

**Step 1: Hindsight reasoning for abduction**: Rewind back to the point where the original action was taken, to infer latent features and noise present in the system at that time. The above counterfactual question, cannot be answered by simply re-running the program with $x = 3$. One must first *abduce* the hidden latent variable $\hat{r} \geq 0.3$ from the observed run $f(\hat{r}, x = 1) = 1$.

**Step 2: Taking a different action (intervention)**: Conditioned on the inferred latent features from abduction stage, perform the counterfactual change by intervening the input to its counterfactual value while keeping everything else the same as in the earlier observation. For the code example, this means holding $\hat{r}$ fixed while intervening by overwriting $x = 3$.

**Step 3: Prediction**: Based on the new action taken, compute its consequences in the counterfactual scenario. In the example, computing $y_{\text{cf}} = f(\hat{r} \geq 0.3, x = 3)$ is final prediction step.

Without latent states and the abduction step, counterfactual reasoning reduces to interventional reasoning, corresponding to Level 2 in Pearl's causal ladder (Pearl, 2009), which breaks down causal reasoning to three progressively more advanced levels: Associational (Level 1), Interventional (Level 2), and Counterfactual (Level 3); see Appendix A for a detailed overview.

**Past studies often overlook abduction.** Prior evaluations of LLM counterfactual reasoning often use fully observed settings with no latent noise. This effectively makes the abduction step unnecessary since there is *no* unobserved variable or noise to abduce. In such regimes, a *counterfactual* query collapses to an *interventional* one: the answer follows directly from taking a different action not requiring the step of inferring hidden state. Take Figure 1 (left) as an example and consider the following question $q$: *What would $y$ have been be if we had set $r = 0.4, x = 3$?* Although $q$ may appear similar to the counterfactual question in Figure 1, it is fundamentally different. Crucially, answering $q$ does not require abducting the values of $r$, since it is explicitly specified. Therefore, solving $q$ relies solely on interventional reasoning (Level 2) rather than counterfactual reasoning (Level 3); in this sense, $q$ effectively collapses to an interventional question despite its seemingly "counterfactual" framing.

The above example question $q$, though synthetic, conceptually illustrates the key reason for the mischaracterization of counterfactuals in many recent works (Wu et al., 2024; Li et al., 2024; Chen et al., 2025; Nguyen et al., 2024b; Paranjape et al., 2022; Wu et al., 2021; Madaan et al., 2021; Ye et al., 2021; Joshi & He, 2022; Vashishtha et al., 2023).[1] See Appendix E for detailed discussions.

Clearly distinguishing counterfactual from interventional reasoning is important for accurately understanding the capabilities and limitations of current LLM paradigms, and for designing algorithms that advance their causal reasoning. It requires an explicit characterization of the three-step process of abduction, intervention, and prediction, which motivates our executable counterfactual framework.

**Other related work.** Jin et al. (2024) provides formal benchmarks across the causal ladder, including counterfactuals. While well grounded in causal theory, some tasks are less aligned with realistic applications and often presuppose familiarity with advanced tools of causal inference (do-calculus, d-separation, mediation, etc.), making it hard to pinpoint whether errors stem from graph inference, identification, effect decomposition, or numerical calculation. Similar trade-offs appear in recent causal benchmarks for LLMs (Yang et al., 2025; Zhou et al., 2024). Operating on code and math, two domains where recent LLMs have made rapid progress, our framework provides concrete mechanisms for isolating their causal capabilities and applying established methods such as SFT and RLVR to enhance counterfactual reasoning (§4). Finally, there is another line of related work on counterfactual example generation for fairness and interpretability research in NLP. We provide a detailed discussion on these topics in Appendix F

## 3 OPERATIONALIZING COUNTERFACTUAL REASONING VIA CODE & MATH

We move beyond graphical approaches (Yang et al., 2025) and purely formal tests (Jin et al., 2023; 2024) by using executable code as an actionable environment for counterfactual reasoning. Because programs are computational graphs, they map naturally onto mathematical and graph formalisms and enable fine-grained control of task difficulty and latent-variable structures. This allows for designing out-of-distribution (OOD) evaluation by encoding causal graphs with novel features and logic unseen during training. Our framework produces executable counterfactuals with verifiable ground-truth outcomes for both evaluation and training.

### 3.1 EXECUTABLE COUNTERFACTUALS: CODE

**Overview.** We generate distinct and executable Python functions from a small set of templates (8 for training, and 3-4 for each evaluation setup) by abstracting out the overall program structure and isolating it from specific variables and operators. Unlike prior work that typically uses a checklist

---

[1]It should be acknowledged that many of these works focus on robustness, generalization, and debiasing, and never intend to study counterfactuals as in the causal sense. Nonetheless, the loose use of the counterfactual framing can lead to misinterpretations by the readers Zhao et al. (2018); Kaushik et al. (2020); Vashishtha et al. (2023), which highlights the importance of a precise characterization of counterfactuals.

```
1  def {function_name}(x, seed_value):
2      random.seed(seed_value)
3      r = random.randint({min_r}, {max_r})
4
5      {preprocessing_block}
6
7      if {condition_type}:
8          if r {comparison_op} {threshold}:
9              result = x {arithmetic_op_1} r
10         else:
11             result = r {arithmetic_op_2} x
12     {elif_block}
13     else:
14         {else_branch_content}
15     return {return_expression}
```

(a) One of the templates for **If_else**
functions in the training dataset.

```
1  def generated_func_1234(x, seed_value):
2      random.seed(seed_value)
3      r = random.randint(2, 8)
4
5      if x > 15:
6          if r < 5:
7              result = x + r
8          else:
9              result = r * x
10     else:
11         result = r - x
12     return result
```

(b) A concrete code function
generated from the template in 2a.

```
1  def generated_func_5007(x, r):
2      if x < 3 or x >= 9:
3          if r >= 4:
4              result = x + r
5          else:
6              result = r + x
7      elif x >= 9:
8          result = r * x + 7
9      elif r >= 4:
10         result = x * 8 * r
11     else:
12         result = r + x
13     return (result * 7 - r) % 18
```

(c) Another structurally different
function that originates from the
same template in 2a.

Figure 2: Structural diversity emerges from our nested template based approach where a single template can generate structurally and semantically different functions as shown in 2b and 2c

approach which merely swaps numbers or operators while keeping the same control flow (Ribeiro et al., 2020), we use function templates where complete code blocks with different functional purposes are replaced by empty placeholders (Figure 2a). Specifically, we apply *Claude-4-Sonnet* to draft these templates and potential code block candidates for each placeholder, and perform manual verification to ensure quality and diversity. For each type of dataset split (training or evaluation) and control logic (if-else, while loop, etc.), we fix a small set of templates along with a list of code block candidates. For training datasets, we supply 15 combinations of function templates and code block candidates. Moreover, to promote finer-grained variations in intermediate computations, we also make operators and variables in the functions changeable. Finally, we deduplicate the generated functions using techniques in Appendix H, which eventually results in a large and diverse set of executable functions using an efficient and controllable recipe.

**Template-based generation.** We consider the following four function logic:

1. **If_else:** These simple functions have at most one level of nesting structure, thus keeping the intermediate computational steps at a low level (Figure 2a).

2. **If_else-long**: To test if the models can generalize to longer code structures with more statements, we construct this evaluation dataset with higher levels of nested if-else structures (Table 9).

3. **While**: To test how models generalize counterfactual reasoning to control logic that it has never seen during training, we construct this dataset with *while loops* (Table 7).

4. **Multi_r**: To test how models generalize to a different causal structure where multiple hidden variables are present, we construct this dataset where each function has three unknown input arguments. Moreover, we level up the complexity by introducing simple *for loops* (Table 8) apart from *if-else* statements.

**If_else** is used for both training and in-distribution (ID) evaluation, while **If_else-long**, **While**, and **Multi_r** are used for out-of-distribution (OOD) evaluation and never used in the training data.

One important feature of our template approach is that there are three different levels of placeholders whose combinations can greatly advance the diversity of our final datasets.

- **Fixed placeholders:** boilerplate such as the function name, a reproducible draw of a latent variable $r$, by setting the random seed, and the final return statement. To design functions with more than one latent variables, we explicitly define placeholders for each extra latent variable.

- **Structural placeholders:** Slots for complete code blocks that define the program's logic, including the optional pre-processing steps, the main `if`-condition (simple or compound), possible `elif` clauses, code pieces inside each branch, and the form of the return statement.

- **Value placeholders:** Specific operators and numbers (e.g., `+`, `*`, thresholds) that determine the function's detailed behavior once the structure is chosen.

To better mirror real-world ambiguity, where multiple latent configurations can explain the same observation, we insert a modulo at the return statement in training functions (i.e., $\text{return } g(\cdot) \bmod m$).

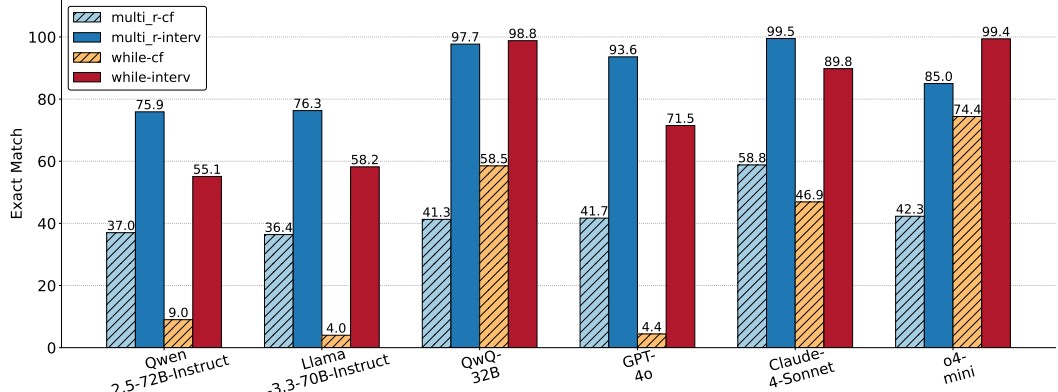

Figure 3: Even for LLMs with strong general capabilities or thinking features, the performance gap between counterfactual and interventional questions originated from the same code function can still be huge, showing the importance of targeted improvements in counterfactual reasoning.

The modulo's periodicity induces a many-to-one mapping from latent $r$ to the observed output, so several $r$ values are consistent with the factual run, yielding multiple valid counterfactual outcomes. At evaluation, we score the model against the full set of valid answers: we report *exact match* (set equality) and an aggregated *F1* that rewards partial coverage of the ground-truth set.

To create the interventional version of the same programming problem, we keep the code unchanged and disclose the realized value(s) of $r$. Revealing $r$ removes the abduction step, so the task reduces to re-evaluating the program under a new input $x$. Please refer to Table 10 for interventional prompt examples.

## 3.2 GSM Math Problem Construction for Counterfactual Reasoning

To test whether models can generalize beyond code, we construct a new dataset of counterfactual variants of GSM-8K-style problems. See Figure 1 (middle) for an illustrative example. The key idea is to introduce a hidden factor in each problem. Taking inspiration from Ye et al. (2024), each problem starts in an everyday setting (office party, school fundraiser, etc.) and is specified by a computational graph that tracks the key quantities (such as counts, unit prices, or fees) and how they combine (sums, percentages, etc.). Inside this graph, we introduce one hidden factor that also contributes to the total; their value is known in the computational graph but not revealed in the narrative. The hidden factors are simple but varied. Examples include: flat add-on (e.g., an unseen service fee), per-item add-on (e.g., an extra fee per tray), and an unknown amount of additional items at a known unit price (e.g., some dessert boxes per tray at $3 each). We verbalize the graph into GSM-style word problems. To increase variety, we use a small set of phrasing templates for different settings (such as office party, fundraiser, etc) and vary both the scenarios and the point where the hidden factor is introduced into the graph. Ground truth answers are produced by executing the computational graphs, therefore resulting in verifiable answers. For creating an interventional version of the problem, we keep everything exactly the same and reveal the value of the latent variable in the problem statement (Table 5). To ensure that the latent variable is used in final answer computation, for each problem constructed, we vary the value of the latent variable and see if it leads to change in final answer. If there is no change we regenerate the problem.

## 4 Experiments

With our executable counterfactuals framework we answer the three research questions in §1.

### 4.1 LLMs Show Weaknesses in Counterfactual Reasoning

**Motivation and Setting.** As discussed in §2, the lack of abduction in prior works reduces counterfactual reasoning to interventional reasoning, thus failing to distinguish the true counterfactual

| Model Class | Model | ID | | OOD | | | | | |
|---|---|---|---|---|---|---|---|---|---|
| | | if_else | | if_else-long | | multi_r | | while | |
| | | F1 | EM | F1 | EM | F1 | EM | F1 | EM |
| Controllably Trained Models | Qwen2.5-1.5B-Instruct | 19.3 | 5.3 | 26.5 | 12.8 | 9.5 | 7.4 | 1.9 | 0.8 |
| | Qwen2.5-1.5B-Instruct-SFT | **62.7** | **44.4** | **51.3** | 32.0 | 21.4 | 20.7 | 2.8 | 2.4 |
| | Qwen2.5-1.5B-Instruct-RL | 34.7 | 20.2 | 50.3 | **39.6** | **25.5** | **25.2** | **5.0** | **4.2** |
| | Qwen2.5-3B-Instruct | 32.1 | 11.8 | 38.7 | 16.7 | 14.0 | 11.1 | 5.4 | 2.7 |
| | Qwen2.5-3B-Instruct-SFT | 70.8 | 53.2 | 55.4 | 34.7 | 22.8 | 21.6 | 2.6 | 2.2 |
| | Qwen2.5-3B-Instruct-RL | **74.8** | **55.2** | **55.9** | **39.3** | **36.9** | **35.9** | **12.9** | **10.5** |
| | Qwen2.5-7B-Instruct | 38.8 | 13.9 | 54.9 | 28.2 | 21.6 | 17.9 | 7.3 | 3.3 |
| | Qwen2.5-7B-Instruct-SFT | 75.8 | 59.0 | 61.4 | 41.7 | 24.9 | 23.3 | 2.5 | 2.1 |
| | Qwen2.5-7B-Instruct-RL | **81.7** | **67.8** | **75.0** | **58.3** | **40.3** | **36.3** | **11.2** | **8.1** |
| General LLMs | Qwen2.5-32B-Instruct | 42.9 | 17.2 | 63.3 | 29.9 | 40.1 | 34.8 | 11.2 | 6.2 |
| | Qwen2.5-72B-Instruct | 47.0 | 20.3 | 65.0 | 32.8 | 42.3 | 37.0 | 13.6 | 9.0 |
| | Llama-3.3-70B-Instruct | 50.0 | 22.0 | 62.8 | 28.7 | 41.8 | 36.4 | 12.0 | 4.0 |
| | GPT-4o | 50.6 | 25.6 | 62.6 | 32.9 | 44.8 | 41.7 | 10.5 | 4.4 |
| | Claude-4-Sonnet | **79.1** | **60.6** | **81.3** | **59.0** | **63.5** | **58.8** | **53.0** | **46.9** |
| Reasoning LLMs | R1-Distill-Qwen-32B | 86.0 | 69.1 | 89.7 | 77.9 | **57.1** | **47.9** | 69.7 | 63.1 |
| | QwQ-32B | 73.5 | 54.9 | 85.1 | 73.0 | 44.7 | 41.3 | 63.2 | 58.5 |
| | o4-mini | **91.1** | **76.2** | **95.9** | **90.2** | 51.9 | 42.3 | **84.6** | **74.4** |

Table 1: Evaluation results on in-distribution (ID) and out-of-distribution (OOD) counterfactual coding tasks using our executable counterfactuals framework. Since each question may contain multiple answers, we report both F1 and exact match scores in percentage units.

from interventional capabilities. In light of this, we pair each of the counterfactual evaluation dataset of our framework with an interventional counterpart, which is built upon the same code function or mathematical conditions except that the originally hidden variable is now revealed and fixed (Table 4 and 10). We evaluate a wide range of models with strong reasoning capabilities and present the comparison results in Figure 3 and Table 5. Please refer to Appendix L.3 for the evaluation hyperparameters adopted throughout this work.

**Findings.** For six strong LLMs spanning four model families in both coding and math domains, there consistently exists a significant performance gap between the counterfactual datasets of our framework and their interventional counterparts, regardless of model providers, sizes, and test-time scaling features. Notably, reasoning models (e.g., QwQ-32B (Team, 2025) and o4-mini) show nearly perfect interventional reasoning performance in coding, yet achieve less than half on counterfactual reasoning. Non-reasoning models mostly score below 10% in counterfactual datasets with while loops, but can achieve over 70% in their interventional counterparts. Therefore, our framework reveals the weakness of current strong LLMs in true counterfactual reasoning, suggesting the necessity of targeted post-training improvements apart from traditional focus on general capabilities only.

## 4.2 Distillation-based SFT Generalizes Poorly

**Motivation and Setting.** We then explore SFT, a widely adopted approach that has been traditionally shown effective for targeted improvements in counterfactual reasoning (Huyuk et al., 2025; Huang et al., 2024). Specifically, we opt for the popular long-Chain-of-Thought (long-CoT) SFT paradigm, where the CoT annotations are distilled by a reasoning model with thinking features, due to its proved benefits of better transfer in reasoning tasks (Guo et al., 2025; Li et al., 2025). We choose *DeepSeek-R1-Distill-Qwen-32B* (Guo et al., 2025) as the teacher model, and *Qwen2.5-1.5B/3B/7B-Instruct* series as the base models for all post-training attempts throughout this work. Please refer to Appendix L.1 for more data annotation and training details.

**Findings.** As shown in Table 1 and Figure 4, compared with their base, SFT models achieve strong in-distribution (ID) counterfactual reasoning performance, as well as decent performance when

certain surface task features (e.g., length of code functions in *if_else-long*) are out-of-distribution (OOD). However, when the fundamental reasoning structures of these tasks become OOD, including the **causal structure** (e.g., more hidden variables in *multi_r*), **control logic** (e.g., while loops as the control structure in *while*), and **question domain** (e.g., from code-based to natural language-based math reasoning in *gsm*), the gains of SFT diminishes and it even hurts the performance in most cases. Thus, our framework demonstrates that long-CoT SFT paradigm has only limited generalization of counterfactual reasoning, despite the powerful external supervision signals. These findings call for investigations into other post-training approaches that are not only more supervision-efficient, but can also generalize to complex and previously unseen task structures.

### 4.3 RLVR ELICITS GENERALIZABLE COUNTERFACTUAL REASONING SKILLS ACROSS CAUSAL STRUCTURES AND QUESTION DOMAINS

**Motivation and Setting.** In search of a supervision-efficient approach to generalize counterfactual reasoning capabilities, we eventually resort to reinforcement learning. We use reinforcement learning from verifiable reward (RLVR) with GRPO (Shao et al., 2024), a popular combination that requires only outcome-based supervision. Following prior work (Sun et al., 2025), we use exact match scores as the outcome-based reward, and set the prompt batch size and rollout size as 16 and 24 respectively. Please refer to Appendix L.2 for more details about RLVR training.

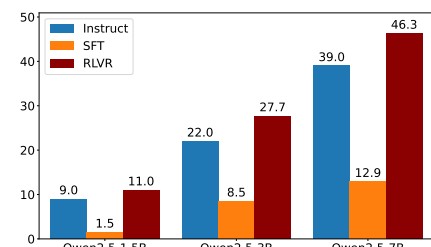

Figure 4: Accuracies on the GSM-counterfactual dataset under domain-transfer. RLVR consistently shows effective generalization from code-based to natural language-based counterfactual reasoning, while SFT consistently fails. Moreover, the improvements of RLVR also robustly scale with the model size.

**Findings.** As shown in Table 1 and Figure 4, RLVR achieves consistent and significant gains for all scales of models, and on all ID and OOD evaluation datasets. The improvements are especially strong on *multi_r*, *while*, and *gsm*, where involve fundamentally OOD causal structures and reasoning contexts, and make our previous SFT attempt uniformly fail. Notably, a *Qwen2.5-7B-Instruct* model trained with RLVR achieves comparable performance with *Qwen2.5-72B-Instruct*, and consistently better performance than its 32B variant across the whole coding domain. Therefore, RLVR successfully achieves our goal of generalizing fundamental counterfactual reasoning skills to complex structures and previously unseen domains with minimal supervision.

## 5 BEHAVIORAL ANALYSIS OF REASONING TRACES

We next analyze the models' reasoning behaviors using executable counterfactuals. Table 2 illustrates the three types of prototypical failure that we observe in the reasoning traces:

1. Brute-force enumeration of all possible hidden-variable values.

2. Assuming an arbitrary value for the hidden variable once the problem is considered too complex.

3. Complicating the problem through unnecessary case splitting and circular analyses.

Inspired by these observations, we evaluate each reasoning trace along two dimensions: *planning* and *execution*. The planning score evaluates whether the three core cognitive skills of counterfactual reasoning—abduction, intervention, and prediction—are sequentially applied. The execution score evaluates the correctness of mathematical computation and code simulation, a general skill that is not specific to counterfactual reasoning. Following prior work (Sun et al., 2025), we use o4-mini as the LLM judge to rate each dimension on a scale of 1 to 5, and defer other technical details, including the grading rubric in prompts, to Table 11 in Appendix. Figure 5 presents the results.

**Scaling model size improves computational accuracy, but not abduction skills.** As shown in Figure 5, across all four coding tasks, scaling up the size of Qwen2.5-Instruct models leads to

| Brute-Force Enumeration | Arbitrary Assumption |
|---|---|
| Since $r$ is not given, list $r$ and unroll loops: $r = 0 \to \texttt{local\_sum} = 1 \to \lfloor 1/3 \rfloor = 0;$ $r = 1 \to \texttt{local\_sum} = 3 \to \lfloor 3/3 \rfloor = 1;$ $r = 2 \to \texttt{local\_sum} = 5 \to \lfloor 5/3 \rfloor = 1;$ . . . scan until $y = 120$ fits at $x = 15$. | Assume $r = 3$, otherwise would be too complex to solve. One outer iter: $\texttt{local\_sum} = 3 + (3 + 1) = 7 \to$ $\lfloor 7/3 \rfloor = 2$. For $x = 12$: $y = 12 \cdot 2 + 12 = 36$. |

| Unnecessary Case-Splitting |
|---|
| Recover $r$ from $x = 15$, $y = 120$ via inner-loop stops. Case 1: $r \leq 0 \to$ inner never runs $\to y = x = 15$. Case 2: $r > 0 \to \texttt{local\_sum}$ after step1 is $r$. Split 2A: stop after step1 if $r \geq 5r$. Split 2B: take step2 if $r < 5r \to \texttt{local\_sum} = 2r + 1$. Also split by $2r + 1$ vs. $5r$ $(<, =, >)$ and by $(2r + 1) \bmod 3 \in \{0, 1, 2\}$; then branch on $q = \lfloor (2r + 1)/3 \rfloor \in \{6, 7, 8\}$ . . . |

Table 2: Examples of three prototypical failure modes in model-generated reasoning traces.

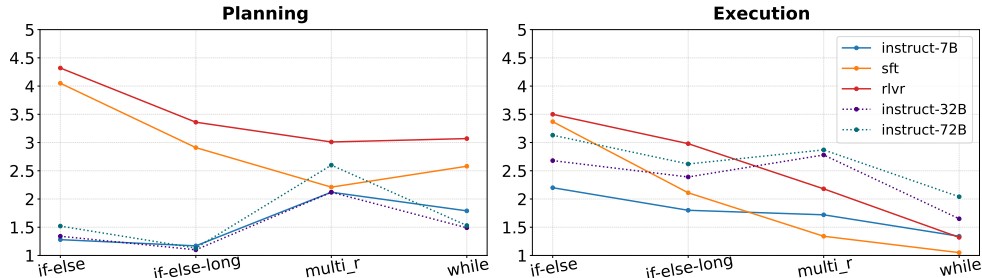

Figure 5: Evaluation results of the LLM-as-a-judge pipeline. For the responses generated by each model on each dataset, the evaluation objective is decoupled into "planning" (left; i.e., whether the "abduction-intervention-prediction" strategy is faithfully followed) and "execution" (right; i.e., whether the intermediate computations are correctly performed).

consistent improvements in execution ratings, but *not* in planning. Instead, the 7B model consistently receives higher ratings for its abduction skills than 32B on 3/4 tasks, and scores even higher than the 72B variant on both *if_else-long* and *while*. This suggests that scaling up the size of LLMs that are post-trained on general domains improves the final accuracy in a way that does not comply with the standard "abduction-intervention-prediction" strategy, thus resulting in poor counterfactual reasoning performance even with a large model size.

**SFT memorizes shallow abduction patterns that fail to generalize to complex problems.** In Figure 5, the planning scores of SFT models substantially drop in OOD tasks. Our inspection of reasoning traces shows that when faced with OOD questions with increased complexity in completing the abduction step, SFT models tend to override the standard reasoning strategy, and instead revert to the prototypical failure modes discussed in Table 2 in order to evade true counterfactual reasoning.

**RLVR generalizes counterfactual reasoning strategies, but is still bottlenecked by computational accuracy.** As Figure 5 also reveals, RLVR models achieve the highest planning scores across all evaluation datasets, demonstrating the generalizable counterfactual reasoning strategy that they learn to apply even in fundamentally OOD tasks. On the other hand, the sharp decrease in execution scores on both *multi_r* and *while* also suggests that a major error type for RLVR is computational errors under the correct reasoning strategy. Therefore, our framework identifies the asynchronism in learning counterfactual reasoning skills and general computational skills, and calls for future efforts into improving both skills simultaneously to build a strong counterfactual reasoning agent.

## 6 CONCLUSION

We address gaps in evaluating counterfactual reasoning in LLMs by decomposing the skill into core components by introducing an executable, code-based framework. Our setup builds dynamic testbeds that require the full abduction, action, prediction rollout and allows for precise control over logic and latent features. Using a template-based approach, we generate many structurally diverse functions to form counterfactual queries and to train smaller models that currently struggle on these tasks. We find that LLMs typically struggle at the abduction step, and this limitation is not resolved by increasing model's size as large scale models (up to 72B) also struggle with this. Our findings show that models trained with SFT transfer these skills in-domain code evaluations but significantly falter on OOD settings, whereas RL consistently induces them from code-only training and generalizes to novel control flows and natural-language counterfactual math. We corroborate this with qualitative case studies and *LLM-as-a-Judge* evaluations. Beyond counterfactuals, the same framework enables flexible evaluation of other causal skills and can help pinpoint where current systems fall short.

## ETHICS AND REPRODUCIBILITY STATEMENT

**Ethics.** All datasets used in our work are created synthetically and will be released publicly. We also make our best efforts provide a fair and thorough comparison between our framework and prior work. To the best of our knowledge, our work investigates fundamental aspects of causal reasoning in the context of large language models, and should not have direct societal impact or implication that should be discussed here specifically.

**Reproducibility.** It is straightforward to construct our framework and reproduce our experimental results, given the implementation details specified in both the main text and appendix. We also plan to release our code and data to further advance the reproducibility of this work.

## ACKNOWLEDGEMENTS

We are grateful to Lifan Yuan, Dylan Zhang, Junlin Yang, Shivam Agarwal, Deema Alnuhait, Daman Arora, Divyat Mahajan, Abhinav Kumar, Kabir Ahuja, Melanie Sclar, Yanai Elazar and members of Chicago Human+AI Lab for their valuable support and insightful discussions. We are also grateful to Prof. Tobias Gerstenberg for the valuable feedback and insights on our idea. This project is partly supported by NSF under award No. 2505932, No. 2126602, an Amazon AICE Award, an award from the Sloan foundation, a grant from Coefficient Giving, gift funding from AI2, and an award from the Open Philanthropy foundation.

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

## LIMITATIONS

One notable limitation of our work is that we focus on evaluating support-set inference instead of full counterfactual distributions. Despite being a limitation, the choice of evaluating support-set inference only was made out of the following reasons:

1. With the form of the latent variable explicitly given (as an input argument $r$ of the code function), support-set inference has exceptional tractability and verifiability.

2. Our results show that frontier models are already failing on support-set inference alone. In light of this, the focus of our work on support-set inference can serve as the first step towards evaluating LLMs on full counterfactual distributions.

3. Furthermore, we would like to note that the majority of the prior works on counterfactual reasoning evaluation, as highlighted in §2, follows a simple setting without the presence of any latent features or randomness, leading to a purely deterministic setting. Therefore, our focus on support-set inference is already a major step towards faithful counterfactual reasoning evaluation.

## A  LLM USAGE STATEMENT

We use Large Language Models for analysis (LLM-as-a-Judge) and data construction in this work, which we have clearly defined in our paper's methodology and analysis sections. LLMs have not been used in our study for other purposes.

## B  CAUSAL LADDER: LEVELS OF CAUSAL REASONING

The seminal work of Pearl (2009) breaks down causal reasoning in three progressively more advanced levels: *Associational* (level 1), *Interventional* (level 2), and *Counterfactual* (level 3):

- **Associational level** concerns observational learning and forms causal hypothesis solely through observations, often interpreted as pattern matching. This mirrors how most machine learning models learn from input features and corresponding labels. This form of learning suffers from potential confounding and selection bias as one cannot perform interventions to identify the underlying causal structure.

- **Interventional learning (level 2)** requires learning through interventions, mirroring how humans typically learn by taking actions and observing the outcomes. While this type of learning might appear to be causal, due to the overall noise in the system which might be changing, identifying whether the observed outcome was solely due to the action performed becomes challenging.

- **Counterfactual reasoning (level 3)** is the highest form of causal reasoning on the causal ladder. It helps in disentangling the effect of other factors in the system, to identify the outcome had the original action been different. However this requires stronger, unit-level structural assumptions, many counterfactuals are not identifiable from data without modeling and this form of reasoning is typically sensitive to model misspecification and "cross-world" assumptions.

| Level | Concept | Expression | Activity | Question | Example |
|---|---|---|---|---|---|
| I | Association / Correlation | $P(y \mid x)$ | Seeing / Observing | How does seeing x change my belief in y? | Would the grass be dry if we found the sprinkler off? |
| II | Intervention / Hypotheticals | $P(y \mid \mathrm{do}(x))$ | Doing | Would y happen if I did x? | Would the grass be dry if we made sure that the sprinkler was off? |
| III | Counterfactuals | $P(y_x \mid x', y')$ | Imagining | Would y have happened instead of $y'$, if I had done $x$ instead of $x'$? / What would have happened if I had done $x$, given that doing $x'$ led to $y'$? | Would the grass have been dry if the sprinkler had been off, given that the grass is wet and the sprinkler on? |

Table 3: Definition of the causal ladder proposed by Pearl (Pearl, 2009), where $\{x, x'\}$ denote candidate causes, $\{y, y'\}$ denote candidate effects, and $P$ denotes the probability of an event. Notably, Interventions and Hypotheticals are different names of the same reasoning paradigm, which only thinks about changes that lie in the **future** (Gerstenberg, 2022). Counterfactuals differs from them by thinking about changes that lie in the observed outcome or **past**.

## C  COUNTERFACTUAL VS INTERVENTIONAL: EXAMPLES OF GSM-BASED TASKS

| Setting | GSM Problem | Answer |
|---|---|---|
| **Counterfactual** | Ravi is organizing an office lunch. Every catering tray is priced at $68. There is also a per-catering tray service fee. A discount of 14% is applied to the items subtotal (before any fees). For 6 catering trays, the total shown is $353.88. If instead 11 catering trays were ordered, with all else unchanged, what amount would be shown? | $648.78 |
| **Interventional** | Ravi is organizing an office lunch. Every catering tray is priced at $68. There is also a per-catering tray service fee of $0.50. A discount of 14% is applied to the items subtotal (before any fees). For 6 catering trays, the total shown is $353.88. If instead 11 catering trays were ordered, with all else unchanged, what amount would be shown? | $648.78 |

Table 4: Two GSM-style instances derived via the dependency-graph approach inspired by Ye et al. (2024). The first row is a *counterfactual* with a hidden latent variable (highlighted) that must be inferred; the second row is the corresponding *interventional* instance with the fee (hidden latent variable) revealed.

# D COUNTERFACTUAL VS INTERVENTIONAL: PERFORMANCE ON GSM-BASED TASKS

| Model | GSM-Interventional | GSM-Counterfactual |
|---|---|---|
| Qwen2.5-1.5B-Instruct | **18.4** | 9.0 |
| Qwen2.5-3B-Instruct | **40.3** | 22.0 |
| Qwen2.5-7B-Instruct | **60.4** | 39.0 |
| Qwen2.5-32B-Instruct | **88.5** | 73.1 |
| Qwen2.5-72B-Instruct | **79.7** | 73.1 |
| Llama-3.3-70B-Instruct | **95.1** | 82.2 |
| GPT-4o | **93.1** | 68.6 |
| DeepSeek-R1-Distill-Qwen-32B | **75.3** | 60.7 |

Table 5: Performance comparison on GSM-based counterfactual and interventional tasks for various models, where the latter is still consistently and significantly higher than the former, echoing prior observation in Figure 3 for code-based tasks.

# E WHEN COUNTERFACTUALS AND INTERVENTIONALS CONFLATE

Wu et al. (2024) analyze GPT-4 under altered premises; since their tasks contain no latent variables, intervention and counterfactual queries coincide, so the reported failures does not probe abductive backtracking effectively. For example one of their evaluated tasks in arithmetic, switching the base from 10 to 9 is simply $\mathrm{do}(\text{base} = 9)$: $27_{10} + 62_{10} = 89_{10}$ but $27_9 + 62_9 = (100)_9$; no latent state needs to be inferred. Consequently, these setups don't diagnose whether a model can perform abduction. Using Fig. 1 as an comparison, the base is $x$ and there is no $r$ in this example. Similar approaches are also adopted in previous works (Li et al., 2024; Nguyen et al., 2024b; Paranjape et al., 2022; Wu et al., 2021; Madaan et al., 2021; Ye et al., 2021; Joshi & He, 2022; Vashishtha et al., 2023). Most of these works use counterfactuals for robustness, debiasing and other purposes. They operate in fully observed settings without latent variability, where the query effectively reduces to an intervention. In contrast, our evaluation targets cases that require abduction, testing whether LLMs can execute the full Abduction-Action-Prediction rollout.

# F ADDITIONAL RELATED WORK

**Causality and LLMs.** Recently a lot of work has focused on how effectively LLMs can be used as domain priors for discovering causal relationship between different real world entities (Kıcıman et al., 2023; Ban et al., 2023; Long et al., 2023; Willig et al., 2023; Vashishtha et al., 2025b). Furthermore, some efforts have also focused on improving LLM's causal reasoning via training on synthetic data (Vashishtha et al., 2025a), or by testing different Chain-of-Thought (CoT) based methods (Jin et al., 2024). Works like Jin et al. (2023; 2024) underline the current limitations of language models' causal reasoning in synthetic and formal settings across different types of reasoning including counterfactual reasoning.

**Using Counterfactuals for NLP tasks.** Past work has also been focusing on improving robustness in NLP tasks such as debiasing for gender-based associations (Wu et al., 2024; Paranjape et al., 2022; Wu et al., 2021; Madaan et al., 2021; Ye et al., 2021; Joshi & He, 2022; Vashishtha et al., 2023), story generation (Qin et al., 2019), fictional complex reasoning (Ahuja et al., 2025), and improving the efficiency of reasoning traces (Lu et al., 2025). Though intended for different purposes, these works uniformly follow a simplified interpretation of counterfactual reasoning (Pearl, 2002b). Another line of research investigates the capabilities of LLMs in generating and evaluating counterfactuals, specifically for data augmentation, fairness enhancement, and interpretability research (Mishra et al., 2024; Nguyen et al., 2024a). However, they focus more on the utility of the generated examples for downstream tasks, instead of a truthful evaluation of counterfactual reasoning itself.

**RL vs SFT.**    Past studies have explored how the training paradigms of SFT and RL based training differ, which guide our training design setup. Kirk et al. (2024) shows how Reinforcement Learning from Human Feedback (RLHF), generalizes better then SFT under distribution shift from train set, however results in lack of diversity. Chu et al. (2025) showed how RL trained on outcome based reward generalizes better across both text and visual tasks, while SFT memorizes the task leading to lack of generalization. However the work emphasises the importance of SFT before RL for effective training. Wu et al. (2025) shows how standard SFT's lack of generalization is due to gradients encoding problematic reward leading to lack of generalization.

We take inspiration from cognitive science literature (Gerstenberg, 2022) to design our program-based analysis in order to evaluate the core cognitive skills required for counterfactual inference. To build math-based generalization tests we build upon the framework of Ye et al. (2024) to generate counterfactual variant of grade school level math problems following a dependency graph based approach. Past work uses programs as world models and simulations, including concept learning from programs (Lake et al., 2015) and code driven or physics based simulators (Cobbe et al., 2020; Freeman et al., 2021) showing the potential of graph for this, which we leverage for our work.

## G    META TEMPLATES: STRUCTURAL PLACEHOLDER DESCRIPTION

| Placeholder | What it controls | Code/line type inserted |
|---|---|---|
| {function_name} | Name of the generated function. | Identifier used in `def` header (snake_case). |
| {min_r}, {max_r} | Bounds for the random draw `r`. | Integer literals or simple expressions inside `random.randint(a,b)`. |
| {preprocessing_block} | Optional setup before branching. | One or more Python statements (e.g., assignments, helper calls). |
| {condition_type} | The top-level `if` condition. | Boolean expression (comparisons, logical ops). |
| {if_branch_content} | Body when `if` is true. | Indented suite: one or more Python statements. |
| {elif_block} | Optional middle branch. | Either empty, or `elif <boolean expr>:` + indented suite. |
| {else_branch_content} | Body when previous conditions are false. | Indented suite: one or more Python statements. |
| {return_expression} | Value the function returns. | Expression used in `return` (identifier, arithmetic expr, tuple, etc.). |

Table 6: Placeholders, roles, and expected code/line types for the if–else meta-template.

## H    DEDUPLICATION AND VERIFICATION OF CODE FUNCTIONS

We validate each function by executing it on a small, randomly generated verification set to ensure it runs without errors.. We also parse the code into Python's Abstract Syntax Tree (AST) to confirm that it compiles without syntax errors. For computing the similarity we convert each generated function into a *structural fingerprint* by counting key elements (if-statements, assignments, operators) and analyzing the overall code pattern. To analyze patterns, it walks through the code structure and identifies sequences like "preprocessing → condition check → branch calculations → return result". It then compares these fingerprints numerically: if two functions have similar counts of each element type and follow the same logical flow pattern, they get a high similarity score $s \in [0, 1]$. Functions with identical structure and execution sequence get a score of $s = 1.0$, while completely different functions score near $s = 0$. Based on manual analysis, we set the threshold at 0.8. This helps identify when the generation process is creating duplicate or overly similar functions that should be filtered out to maintain training data diversity.

# I COUNTERFACTUAL REASONING PROMPTS FOR CODE FUNCTIONS

## I.1 WHILE

---

*You are a language model that reasons about code without using any external execution environment. Do not simply repeat the prompt. Instead, analyze the Python function below, provide step-by-step reasoning, and answer the counterfactual question.*

*Python function:*

```python
def generated_func_997660_100(x, r):

    primary_sum = 0
    secondary_sum = 0
    counter = 0

    while counter < x:
        primary_sum += r + counter
        secondary_sum += counter * 2

        if primary_sum > secondary_sum:
            primary_sum -= 5

        counter += 1

    return (primary_sum + secondary_sum) // 5
```

*Observed call:*
When this function was called with input $x = 10$, it produced the output $y = 36$.

*Counterfactual query:*
If instead of $x = 10$, we had called this function with a different input value of $x = 8$ while keeping everything else unchanged, what could the output $y$ have been? Let's think step by step to get the answer.

*Required answer format:*
\boxed{ans1, ans2, ans3}

---

Table 7: Counterfactual prompt example for *While* dataset.

## I.2 MULTI_R

*You are a language model that reasons about code without using any external execution environment. Do not simply repeat the prompt. Instead, analyze the Python function below, provide step-by-step reasoning, and answer the counterfactual question.*

*Python function:*

```python
import random

def generated_func_1136(x, r1, r2, r3):

    prep = x * (r2 + r3)

    if x == r1:
        result = x * r3
        for i in range(2):
            pass
        result = result = x + r2
    else:
        result = x - r2
        for j in range(6):
            pass
        result = result = x + r1

    result = result * (r1 + r2 * r3)
    return result
```

*Observed call:*
*When this function was called with input $x = 16$, it produced the output $y = 3640$.*

*Counterfactual query:*
*If instead of $x = 16$, we had called this function with a different input value of $x = 18$ while keeping everything else unchanged, what could the output $y$ have been? Let's think step by step to get the answer.*

*Required answer format:*
```
\boxed{ans1, ans2, ans3}
```

Table 8: Counterfactual prompt example for *Multi_r* dataset

## I.3 IF_ELSE-LONG

---

*You are a language model that reasons about code without using any external execution environment. Do not simply repeat the prompt. Instead, analyze the Python function below, provide step-by-step reasoning, and answer the counterfactual question.*

*Python function:*

```python
def generated_func_1194(x, r):
    alt4 = 10
    final2 = 1
    final3 = 0
    final4 = 4
    temp1 = 3
    temp2 = 3
    temp3 = 2
    r = abs(r)

    if r > 9:
        temp1 = (x % 1) - (r % 10)
        if (r % 10) == 5:

            if temp1 < 5:

                if (temp3 * x) < r:
                    final4 = (temp3 * x) + 2
                    result = final4 + x
                else:
                    alt4 = x - temp3
                    result = alt4 + r
            else:
                final3 = temp2 + r
                result = final3 - x
        else:
            final2 = (temp1 ** 5) * r
            result = final2 * r
    else:
        else_val = (r ** 4) * x
        result = else_val + r

    return result % 6
```

*Observed call:*
When this function was called with input $x = 18$, it produced the output $y = 4$.

*Counterfactual query:*
If instead of $x = 18$, we had called this function with a different input value of $x = 20$ while keeping everything else unchanged, what could the output $y$ have been? Let's think step by step to get the answer.

*Required answer format:*
\boxed{ans1, ans2, ans3}

---

Table 9: Counterfactual prompt example for *If_else-long* dataset.

## J  INTERVENTIONAL REASONING PROMPTS FOR CODE FUNCTIONS

*You are a language model that reasons about code without using any external execution environment. Do not simply repeat the prompt. Instead, analyze the Python function below, provide step-by-step reasoning, and answer the **interventional** question.*

**Python function:**

```python
def generated_func_1273(x, r1, r2, r3):

    prep = x + (r2 - r3)

    if x != r1:
        result = x + r2
        for i in range(6):
            pass
        result = result = x * r1
    else:
        result = x + r3
        for j in range(2):
            pass
        result = result = x + r3

    result = result + (r1 - r2 - r3)
    return result
```

**Observed call:**
*When this function was called with inputs $x = 18$, $r_1 = 20$, $r_2 = 5$, and $r_3 = 17$, it produced the output $y = 358$.*

**Interventional query:**
*If instead of $x = 18$, we had called this function with $x = 20$ while keeping $r_1 = 20$, $r_2 = 5$, and $r_3 = 17$ unchanged, what* could *the output $y$ have been? Let's think step by step to get the answer.*

**Required answer format:**
`\boxed{ans1, ans2, ans3}`

Table 10: Interventional prompt example for *Multi_r* dataset. We omit the interventional examples of the remaining three code-based datasets, as their prompt templates are mostly similar to this one, and the examples of python functions are already displayed in Appendix I

## K   PROMPT TEMPLATE FOR LLM-AS-A-JUDGE ANALYSIS

---

*You are presented with a counterfactual reasoning question about a code function, along with a sample solution. Your task is to carefully analyze this solution and rate how it performs in terms of planning and execution, on a scale from 1 to 5.*

***Criteria for rating planning:***
***5** – The solution adopts a perfect plan for all such counterfactual questions with two stages: (1)* Backward Reasoning with Original Data*: determine the value(s) of the unknown variable $r$ by setting up a mathematical equation based on the arithmetic operations performed on $r$ and the original input $x$ within the code path that produced the original output $y$. (2)* Forward Reasoning with Counterfactual Data*: use the value(s) of $r$ found in the previous step to determine the new output(s) based on the counterfactual input.*
***3** – The solution shows awareness of first finding values of $r$ from the original $x$ and $y$, and then computing the new outputs using the same $r$, but it does not follow this Backward-then-Forward plan faithfully or decisively. For instance, it hesitates about solvability without an explicit $r$, or resorts to brute-force enumeration without persisting in the desired plan.*
***1** – The solution does not align with the Backward-then-Forward plan at all (e.g., starts with brute-force enumeration of $r$ without using the given $x$, $y$ to determine $r$ smartly).*

***Criteria for rating execution:***
*Score execution based on whether the solution follows the code-simulation paths and performs step-by-step numerical computations faithfully and correctly. More simulation/computation mistakes $\rightarrow$ lower execution score.*

***Question:***
`{prompt}`

***Ground Truth Answers:***
`{ground_truth}`

***Solution:***
`{response}`

***Required response format (JSON):***
```json
{
    "planning": [1|2|3|4|5],
    "planning_explanation": "first briefly describe the
    planning or strategy this solution adopts, and then explain
     why you gave this planning score",
    "execution": [1|2|3|4|5],
    "execution_explanation": "brief explanation of why you gave
     this execution score"
}
```

---

Table 11: The prompt template for LLM-as-a-judge analyses in §5, with detailed evaluation rubrics for both **planning** and **execution** scores.

## L   TRAINING AND EVALUATION DETAILS

### L.1   SFT TRAINING SETUPS

**Reasoning Trace Generation.**   Our training dataset is built upon 5500 code-based counterfactual reasoning prompts that only involve *if_else* logic. We leverage *DeepSeek-Distilled-Qwen-32B-Instruct* Guo et al. (2025) to annotate the reasoning traces for these prompts through rejection sampling Li et al. (2025). Specifically, for each prompt, we sample multiple responses until the model reaches the correct final answer or a budget of $N = 8$ is reached. The resulting training set achieves a final F1 score of 95.9 and exact match score of 88.9, ensuring the correctness of SFT training signals.

**Training Configurations.** Throughout this work, we follow prior practice (Ye et al., 2025) by performing full-parameter SFT training using the LlamaFactory framework (Zheng et al., 2024). All the SFT experiments are carried out using four NVIDIA H100 GPUs, with DeepSpeed Zero-3 (Rasley et al., 2020), FlashAttention-V2 (Dao et al., 2022), and Liger Kernel (Hsu et al., 2025) enabled to improve time and memory efficiency. The key training hyperparameters are shown below:

```
--cutoff_len 16384
--num_train_epochs 2
--bf16 True
--optim adamw_torch
--lr_scheduler_type cosine
--learning_rate 5e-05
--warmup_ratio 0.05
--weight_decay 0.0
--per_device_train_batch_size 1
--gradient_accumulation_steps 4
--seed 42
```

### L.2 RL TRAINING SETUPS

Throughout this work, we follow prior practice (Luo et al., 2025; Sun et al., 2025) by performing full-parameter RLVR training with the verl (Sheng et al., 2024) framework. We use four NVIDIA H100 GPUs for training 1.5B and 3B models, and eight H100 GPUs for 7B models. We adopt an effective prompt batch size of 16, a rollout batch size of 24, a prompt length limit of 512, a response length limit of 2000, a sampling temperature of 1.0, a coefficient of $10^{-3}$ for low-variance KL auxiliary loss, and a total of 1500 training steps. The reward is the simple exact match score.

### L.3 EVALUATION SETUPS

**Dataset Statistics.** The in-distribution (ID) *If_else* evaluation dataset contains 500 examples, while the three out-of-distribution (OOD) evaluation datasets, *If_else-long*, *Multi_r* and *While*, contain 480, 575 and 480 examples respectively.

**Evaluation Protocol.** Throughout this work, we use the vLLM framework (Kwon et al., 2023) for efficient evaluation. Specifically, we follow prior practice (Luo et al., 2025; Guo et al., 2025) by using sampling with a temperature of 0.6, a top-p of 0.95, and a maximum of 16000 generated tokens to generate $k = 3$ responses per question. For each evaluation dataset, we report the average accuracy over $k$ responses (i.e., avg@k) to reduce the variance in performance statistics.

## M BENCHMARKS WITH RICHER CAUSAL STRUCTURES: *x_determines_r*

To further enrich the diversity of causal structures within our benchmark, we introduce a new task type: *x_determines_r*. Unlike previous settings where latent variables are independent of the intervention variable, this task explicitly specifies a dependency between $x$ and $r$ from the onset. Specifically, the value of $r$ is sampled from a Gaussian distribution where the mean and standard deviation are computed functions of $x$. This construction creates a mediator effect ($x \to r \to y$) in addition to the direct effect of $x$ on $y$.

Table 12 presents an example of a function from this dataset. The implementation defines parameters $\mu$ and $\sigma$ based on the input $x$, and subsequently samples $r$. This dataset comprises 600 examples in the code domain.

We evaluated Qwen2.5-1.5B-Instruct, its SFT and RLVR variants, and proprietary LLMs (GPT-4o and Claude-3.5-Sonnet) on this new dataset. The results are presented in Table 13. We observe a trend consistent with the main findings in Table 1: RL demonstrates superior generalization performance on this new causal structure involving dependencies between $x$ and $r$. Notably, the small RL-trained model exceeds the performance of GPT-4o by a significant margin in this setting.

```
def generated_func_gauss_6230(x, seed_value):
    random.seed(seed_value)

    min_r = -50
    max_r = 50
    mu = x * 1.08 + 6.68
    sigma = max(1, abs(x) * 0.08 + 3.47)

    r = int(round(mu + sigma * random.gauss(0, 1)))
    r = max(min_r, min(max_r, r))

    if x >= 14:
        result = (x - r) * 3
    else:
        if r <= x - 6:
            result = (x + r) // 2 + 3
        else:
            result = (r - x) + 5

    return result if result > 4 else (r - x)
```

Table 12: An example code function from the *x_determines_r* dataset, where the latent variable $r$ acts as a mediator dependent on $x$.

| Model | F1 |
|---|---|
| Qwen2.5-1.5B-Instruct | 24.5 |
| Qwen2.5-1.5B-Instruct-SFT | 45.6 |
| Qwen2.5-1.5B-Instruct-RL | **58.2** |
| Claude-3.5-Sonnet | **85.1** |
| GPT-4o | 44.0 |

Table 13: Evaluation results on the *x_determines_r* dataset. The RL-trained model demonstrates stronger generalization to the mediator causal structure compared to SFT and GPT-4o.

## N  MULTI-JUDGE VALIDATION OF LLM-AS-A-JUDGE ANALYSIS

To solidify the reliability of the planning and execution scores used in our analysis, we performed both human validation and multi-judge comparison to ensure the metrics reflect genuine reasoning quality rather than stylistic variance.

Specifically, we uniformly sampled 30 examples from the in-distribution task (*if_else*) and 30 examples from the out-of-distribution task (*multi_r*), for a total of 60 examples. We independently applied both manual human scoring and model-based scoring using Claude-4.5-Sonnet, while maintaining the exact same grading rubric used for the o4-mini judge in §5.

We calculated the Pearson correlation between each pair of the three rating subjects: (1) Human, (2) o4-mini, and (3) Claude-4.5-Sonnet. Tables 14 and 15 present the correlation matrices for planning and execution scores, respectively. The consistently high correlations suggest strong alignment between human and automated judges, validating the generality and robustness of the behavioral analysis reported in §5.

Table 14: Pearson correlation matrix for **Planning** scores.

|  | Human | o4-mini | Sonnet |
|---|---|---|---|
| **Human** | 1.00 | 0.76 | 0.63 |
| **o4-mini** | 0.76 | 1.00 | 0.80 |
| **4.5-Sonnet** | 0.63 | 0.80 | 1.00 |

Table 15: Pearson correlation matrix for **Execution** scores.

|  | Human | o4-mini | Sonnet |
|---|---|---|---|
| **Human** | 1.00 | 0.86 | 0.74 |
| **o4-mini** | 0.86 | 1.00 | 0.80 |
| **4.5-Sonnet** | 0.74 | 0.80 | 1.00 |

