# OpenReview forum: "Executable Counterfactuals: Improving LLMs' Causal Reasoning Through Code"
_ICLR.cc/2026/Conference — ICLR 2026 Poster_

### Official Review · Reviewer_yTEV · 2025-10-24

**Soundness:** 3
**Presentation:** 3
**Contribution:** 3
**Rating:** 8
**Confidence:** 3

**Summary:**

The paper proposes a new benchmark for evaluating counterfactual reasoning in large language models. By framing tasks as executable code and math problems, it tests whether models can perform the full causal reasoning cycle (abduction, intervention, and prediction). Using this setup, the authors find that LLMs, regardless of size, struggle especially with the abduction step. They introduce an LLM-as-a-judge method to rate reasoning quality in terms of planning and execution, revealing that reinforcement learning with verifiable rewards (RLVR) induces more consistent causal reasoning than supervised fine-tuning (SFT). The work provides a structured framework for diagnosing reasoning failures in LLMs.

**Strengths:**

The paper is original in defining executable counterfactuals, i.e., a new, code-based benchmark that captures causal reasoning process (abduction, intervention, prediction). The technical quality is strong, combining formal causal modeling with large-scale experiments comparing model types and training methods (SFT vs. RL).  In terms of clarity, the presentation is clear and well-structured, using intuitive examples. The significance comes from establishing a scalable framework for testing reasoning in LLMs.

**Weaknesses:**

The evaluation is limited to synthetic and code-based tasks, leaving unclear how the framework extends to real-world reasoning. Adding at least one natural dataset or human-grounded task would strengthen generalizability.

The LLM-as-a-judge lacks calibration against human evaluators. Without inter-rater validation or multi-judge comparison, reliability of the planning and execution scores remains uncertain.

**Questions:**

1. How could the proposed framework be adapted to naturalistic or real-world reasoning tasks? Any datasets/benchmarks that could be added?

2. How did the authors validate the accuracy and consistency of the o4-mini LLM-as-a-judge beyond rubric standardization? Was any human–LLM agreement test or multi-judge comparison performed to confirm that planning and execution scores reflect genuine reasoning quality rather than stylistic variance?

3. Any intuition about which parts of the reinforcement pipeline are responsible for the improved results, i.e., the emergence of structured reasoning?

---

> ### Author Response · Authors · 2025-11-22
> **Response to Reviewer yTEV (1/2)**
>
> Thank you for your informative feedback and insightful suggestions! We are encouraged that the reviewer acknowledges the originality, technical quality, and significance of our work. We are also glad that our presentation is regarded as clear and well-structured. Below we respond to your questions and concerns.
>
>
> > **1. The evaluation is limited to synthetic and code-based tasks, leaving unclear how the framework extends to real-world reasoning. How could the proposed framework be adapted to naturalistic or real-world reasoning tasks? Any datasets/benchmarks that could be added?**
>
> We thank the reviewer for this point. While we agree that our setup is synthetic and code-based, **this choice is deliberate: it allows us to exactly specify the structural equations and enumerate all possible counterfactual outcomes for each instance.** In real-world scenarios, estimating counterfactuals requires identifying all relevant latent variables and accurately modeling substantial, unobserved noise—assumptions that are barely verifiable in practice. To precisely assess how well models perform abduction, a step that has not been explicitly evaluated in prior work, we therefore adopt a controlled executable setting where the ground-truth latent space and counterfactual outcomes are fully known.
>
> Building realistic counterfactual benchmarks remains challenging because of this unavoidable uncertainty over environment variables, but **we propose that one can move toward greater realism by increasing the complexity of the underlying code (e.g., richer dynamics, stochastic branches) and injecting additional noise**. For example, [1] models predictable human behavioral “scripts” as executable programs and uses LLMs and probabilistic inference to recover a hypothesis space of such programs in constrained grid-world and embodied tasks. A similar code-as-environment design could be adapted to construct more realistic counterfactual benchmarks, while still operating in a controlled setting that makes rigorous evaluation feasible. Given the limited scope of a single research paper, we leave the direction of increasing the complexity of underlying code to better represent the read world to future work.
>
>
> > **2. The LLM-as-a-judge lacks calibration against human evaluators. Without inter-rater validation or multi-judge comparison, reliability of the planning and execution scores remains uncertain. How did the authors validate the accuracy and consistency of the o4-mini LLM-as-a-judge beyond rubric standardization? Was any human–LLM agreement test or multi-judge comparison performed to confirm that planning and execution scores reflect genuine reasoning quality rather than stylistic variance?**
>
> We thank the reviewer for this valuable feedback. In order to solidify the reliability of planning and execution scores, we perform both human validation and multi-judge comparison.
>
> Specifically, we uniformly sample 30 examples from an in-distribution task (i.e., `if_else`) and an OOD task (e.g., `multi_r`) respectively. For these 60 examples altogether, we independently apply both manual scoring and Claude-4.5-Sonnet-based scoring, while keeping the grading rubric unchanged in the whole process. We then calculate the Pearson correlation between each pair of the three rating subjects, (1) human, (2) o4-mini, and (3) 4.5-Sonnet, over the 60 examples, and present the correlation matrices below for planning and execution scores respectively. The overall high correlations suggest strong alignment among human and LLM judges, thus validating the generality and robustness of our behavioral analysis in Section 5.
>
>
> |  **Planning**  | **human** | **o4-mini** | **4.5-Sonnet** |
> |:--------------:|:---------:|:-----------:|:--------------:|
> |    **human**   |    1.00   |     0.76    |      0.63      |
> |   **o4-mini**  |    0.76   |     1.00    |      0.80      |
> | **4.5-Sonnet** |    0.63   |     0.80    |      1.00      |
>
>
> | **Execution**  | **human** | **o4-mini** | **4.5-Sonnet** |
> |:--------------:|:---------:|:-----------:|:--------------:|
> | **human**      | 1.00      | 0.86        | 0.74           |
> | **o4-mini**    | 0.86      | 1.00        | 0.80           |
> | **4.5-Sonnet** | 0.74      | 0.80        | 1.00           |

---

> > ### Author Response · Authors · 2025-11-22
> > **Response to Reviewer yTEV (2/2)**
> >
> > > **3. Any intuition about which parts of the reinforcement pipeline are responsible for the improved results, i.e., the emergence of structured reasoning?**
> >
> > Thanks for raising this interesting question! In our engineering practice, we find increasing the extent of exploration (i.e., temperature, prompt batch size, and rollout batch size) is especially helpful for the improvement in reward and generalized reasoning performance of RLVR. This observation echoes prior works [2,3,4] that systematically establish the importance of promoting exploration and diversity in the online sampling process of RLVR. There has also been an ongoing debate in the community on how the online and on-policy nature of RLVR [5,6] can be the key to boost generalization in complex reasoning, which we think is also likely to account for the improved reasoning generalization in our case.
> >
> > [1] Jha et al. Modeling Others' Minds as Code. arXiv:2510.01272.
> >
> > [2] Agarwal et al. The Unreasonable Effectiveness of Entropy Minimization in LLM Reasoning. NeurIPS 2025.
> >
> > [3] Wu et al. The Invisible Leash: Why RLVR May or May Not Escape Its Origin. 	arXiv:2507.14843.
> >
> > [4] Liu et al. Scaling Up RL: Unlocking Diverse Reasoning in LLMs via Prolonged Training. 	arXiv:2507.12507.
> >
> > [5] Shenfield et al. RL's Razor: Why Online Reinforcement Learning Forgets Less. 	arXiv:2509.04259.
> >
> > [6] Lv et al. Towards a Unified View of Large Language Model Post-Training. 	arXiv:2509.04419.

---

> ### Comment · Reviewer_yTEV · 2025-11-26
> **I thank the authors for the clarifications. I have no further questions. I believe my initial score is fair.**
>
> I thank the authors for the clarifications. I have no further questions. I believe my initial score is fair.

---

### Official Review · Reviewer_V1ez · 2025-10-30

**Soundness:** 3
**Presentation:** 2
**Contribution:** 2
**Rating:** 4
**Confidence:** 4

**Summary:**

This paper highlights that current LLMs perform poorly in identifying counterfactuals within coding tasks. Moreover, the effectiveness of supervised fine-tuning (SFT) remains limited, as it fails to generalize to unseen scenarios. Finally, the authors demonstrate that reinforcement learning (RL) achieves strong performance and shows promising potential for generating counterfactual examples.

**Strengths:**

Strength:
1 The topic is interesting.
2 The experiments are abundant.
3 The introduction of the template-based generation approach is clear and concise. The inclusion of a concrete example effectively clarifies the goal of the task.

**Weaknesses:**

Weakness:
1 Although this paper demonstrates the differences among algorithms in counterfactual reasoning, it fails to provide an in-depth analysis of the observed phenomena. In other words, the work reads more like an experimental report than a research paper. For instance, the authors claim that reinforcement learning (RL) exhibits strong generalization ability, yet offer no explanation or supporting analysis for this claim.
2 The paper lacks a clear definition of counterfactual examples, which makes it difficult to understand the exact task at first. In fact, the task only becomes clear upon reading Section 3.

**Questions:**

Questions and Suggestions:
1 I suggest that the authors provide additional analysis or, if possible, a theoretical guarantee to better explain and support their findings.
2 I recommend adding a preliminary section before Section 3 to clearly define the problem and formalize the task setup.
3 The paper would also benefit from including a discussion of previous work on counterfactual example generation. I list several relevant papers below for reference:
1 Mishra, Ashish, Gyanaranjan Nayak, Suparna Bhattacharya, Tarun Kumar, Arpit Shah, and Martin Foltin. "Llm-guided counterfactual data generation for fairer ai." In Companion Proceedings of the ACM Web Conference 2024, pp. 1538-1545. 2024.
2 Nguyen, Van Bach, Paul Youssef, Christin Seifert, and Jörg Schlötterer. "Llms for generating and evaluating counterfactuals: A comprehensive study." arXiv preprint arXiv:2405.00722 (2024).

---

> ### Author Response · Authors · 2025-11-22
> **Response to Reviewer V1ez (1/1)**
>
> Thank you for your informative feedback and suggestions! We are encouraged that the reviewer finds the topic of our work interesting, considers the experiments in our work abundant, and also regards our template-based data generation approach as clear and concise. Below we respond to your questions and concerns.
>
> > **1. Although this paper demonstrates the differences among algorithms in counterfactual reasoning, it fails to provide an in-depth analysis of the observed phenomena. For instance, the authors claim that reinforcement learning (RL) exhibits strong generalization ability, yet offer no explanation or supporting analysis for this claim. I suggest that the authors provide additional analysis or, if possible, a theoretical guarantee to better explain and support their findings.**
>
> Thank you for this suggestion!
>
> 1. We want to first clarify that **in Section 5, we have provided detailed behavioral analyses** on the counterfactual reasoning traces generated by models trained with SFT and RLVR. Through in-depth manual analyses (Table 2) and LLM-as-a-judge ratings (Figure 5; which has also been augmented by alignment study with human judges. Please see our response to Reviewer yTEV for more details), we show that SFT models mostly memorize shallow abduction patterns that cannot generalize to complex OOD problems, and tend to revert to the three prototypical failure modes displayed in Table 2 in order to evade true counterfactual reasoning. Moreover, we also show that RLVR achieves strong performance by generalizing the “abduction-intervention-prediction” strategy to fundamentally OOD setups. We believe these analyses offer solid explanations for the findings on our benchmark.
>
> 2. While we appreciate the reviewer’s suggestion on adding theoretical explanations to further support our findings, we would like to note that this does not closely align with the focus of our work. More specifically, the aim of this work is to propose **the importance of verifiable counterfactual reasoning with explicit abduction, and the potential of RLVR as a preliminary solution**, both of which have been highlighted by our empirical findings in Section 4 and 5. It would be out of our scope to discuss the theoretical factors that guarantee the generalization of RL in counterfactual reasoning. **Nevertheless, we have included a discussion on the generalization properties of RL and SFT in Appendix F (i.e., Appendix E in the original submission; same below), and also recommend referring to our response to Reviewer yTEV**, where we include a review of past literature that discusses the components of the RLVR pipeline that contribute most to its generalization success. We hope it can further solidify the confidence in our observations and analyses.
>
>
> > **2. The paper lacks a clear definition of counterfactual examples, which makes it difficult to understand the exact task at first. In fact, the task only becomes clear upon reading Section 3. I recommend adding a preliminary section before Section 3 to clearly define the problem and formalize the task setup.**
>
> We thank the reviewer for this valuable feedback. We would like to first clarify that in Section 1 (Line 97 - 102), we provide an initial formalization of our counterfactual reasoning task setup by referencing the illustrated example in Figure 1. In Section 2 (Line 140 - 156), we further provide a detailed definition of the true counterfactual reasoning that we aim to evaluate by specifying the three required cognitive steps. In our paper revision, we will make this connection more explicit and expand the description of the task structure in the introduction, so that readers can more easily understand the overall task setups.
>
>
> > **3. The paper would also benefit from including a discussion of previous work on counterfactual example generation.**
>
> Thank you for the useful feedback! In Appendix F, we actually cover an initial discussion on the generation and applications of counterfactuals in NLP. In the updated PDF revision, we will make sure to further augment the discussion on counterfactual example generation in both Section 2 and Appendix F, with the incorporation of the works referenced by the reviewer.

---

> > ### Author Response · Authors · 2025-11-27
> > **Followup on Rebuttal**
> >
> > Dear Reviewer V1ez,
> >
> > Thank you again for your thoughtful review of our paper. If there are any remaining concerns or points where additional clarification from our side would help your assessment, we would be happy to provide further details.

---

### Official Review · Reviewer_ZkHM · 2025-11-01

**Soundness:** 3
**Presentation:** 3
**Contribution:** 2
**Rating:** 6
**Confidence:** 3

**Summary:**

The paper investigates whether large language models (LLMs) can perform counterfactual reasoning, which requires the causal sequence of abduction, intervention, and prediction. The authors argue that existing LLM evaluations on counterfactual reasoning often overlook the abduction step, inferring hidden latent variables from factual observations, which is essential in Pearl’s framework.

To make abduction explicit and verifiable, the paper introduces a counterfactual reasoning benchmark in the form of code-based tasks. Each task corresponds to a function $Y = f(X, R_1, R_2, \ldots)$ where $X$ is the input and $R$ represents latent variables sampled inside the function. Given factual observations $X = x$ and $Y = y$, the goal is to infer the support set of the counterfactual $Y$ had $X$ been $x’$. Experiments with open-source models (1.5B–72B) and commercial reasoning models reveal that they consistently fail at counterfactual reasoning due to an inability to perform abduction.

Next, the paper studies whether fine-tuning can induce counterfactual reasoning. Two approaches are compared: Supervised Fine-Tuning (SFT) and Reinforcement Learning with Verifiable Reward (RLVR). Both methods enable strong performance on in-distribution code counterfactual tasks. However, RLVR generalizes significantly better, to functions $f$ with unseen structural patterns and to a natural language counterfactual dataset (GSM8K-style math word problems constructed from causal graphs), while SFT collapses to near-zero performance.

**Strengths:**

**S1**. The paper highlights the crucial role of inferring latent variables R (abduction) in counterfactual reasoning. This step is often ignored in existing LLM counterfactual evaluations. The paper provides clear motivation and concrete illustrations for why abduction must be explicitly evaluated, and how current methods implicitly avoid it.


**S2**. The proposed code-based benchmark is both systematic and innovative. By embedding latent randomness into executable coding functions, the authors create a counterfactual reasoning task with verifiable ground truth. The methodology can offer insights that could be extended to broader counterfactual reasoning tasks.

**Weaknesses:**

The definition of counterfactual reasoning used in the paper is narrower than the standard understanding in causal inference.

**W1**. In Pearl’s framework, abduction requires inferring the posterior distribution $P(R \mid x, y)$, followed by computing the counterfactual distribution $P(Y_{x'} | x, y) = \sum_R P(Y_{x'} | r)P(r \mid x, y)$. In contrast, the benchmark in this paper focuses on identifying the **support set** of latent variables consistent with the observation, and then predicting the **support set** of counterfactual outcomes. It is suggested to clarify that the benchmark evaluates support-set inference, not full counterfactual distributions.

**W2**.
Although the paper claims to move beyond graphical approaches, every task ultimately reduces to the functional form $Y = f(X, R_1, R_2, \ldots, R_n)$, without causal dependencies between $X$ and $R$ and other observed variables. As a result, the causal graph collapses to a simple structure $X \rightarrow Y$. Does the benchmark proposed in the work include any tasks with richer causal structures, either in the code setting or the natural language setting?

**Questions:**

**Q1**. In line 101, the paper states: “infer r based on the observation $y = -1$ (abduction).”
Should this instead be “infer $r$ based on the observation $x = 1, y = -1$”?
Also, does abduction always consider both x and y as observations in this benchmark?

**Q2**. See **W2**

---

> ### Author Response · Authors · 2025-11-22
> **Response to Reviewer ZkHM (1/1)**
>
> Thank you for your insightful feedback and useful suggestions! We are encouraged that the reviewer acknowledges the crucial role that our evaluation focus, the inference of latent variables, plays in counterfactual reasoning. We are also glad to know that the reviewer finds our code-based framework systematic, innovative, and has the potential to be extended to broader counterfactual reasoning tasks. Below we respond to your questions and concerns.
>
>
> > **1. It is suggested to clarify that the benchmark evaluates support-set inference, not full counterfactual distributions**
>
> We thank the reviewer for this valuable suggestion. We agree that our benchmark focuses on evaluating support-set inference instead of full counterfactual distributions, and we will make sure this detail is emphasized in the updated revision for clarity. We would also like to clarify the reasons for this choice:
>
> 1. We settled down on the evaluation of support-set inference due to its tractability and verifiability, with the form of the latent variable R explicitly given.
>
> 2. Our results show that frontier models are failing on support-set inference alone. In light of this, the focus of our work on support-set inference can serve as the first step towards evaluating LLMs on full counterfactual distributions.
>
> 3. Furthermore, we would like to note that the majority of the prior works on counterfactual reasoning evaluation, as highlighted in Section 2, follows a simple setting without the presence of any latent features or randomness, leading to a purely deterministic setting. Therefore, our focus on support-set inference is already a major step towards faithful counterfactual reasoning evaluation. Narrowing down the scope of evaluation does not hurt our originality, but instead solidifies the verifiability of our benchmark.
>
> We will add a “Limitations” section in our paper to augment the discussion on this point.
>
>
>
> > **2. Does the benchmark include any tasks with richer causal structures (such as dependencies between $X$ and $R$)?**
>
> We thank the reviewer for raising this insightful question. We want to first clarify that due to the richness of code semantics, different types of executable counterfactual tasks actually have different graphical structures even though they share the same functional form  $Y = f(X, R_1, R_2, \ldots, R_n)$. For example, the loop semantics in the `while` task involves iterative and interleaved operations between X and R, thus differing its causal structure from the simpler branching semantics in `if_else`.
>
> To further enrich the causal structures contained in our benchmark, we also create a new type of task, `x_determines_r`, which explicitly specifies dependency between X and R from the very beginning, as it samples the value of R from a gaussian distribution whose mean and standard deviation is computed using the value of X, therefore creating a mediator effect between X and Y using R.
>
> ```python
> def generated_func_gauss_6230(x, seed_value):
>     random.seed(seed_value)
>
>     min_r = -50
>     max_r = 50
>     mu = x * 1.08 + 6.68
>     sigma = max(1, abs(x) * 0.08 + 3.47)
>
>     r = int(round(mu + sigma * random.gauss(0, 1)))
>     r = max(min_r, min(max_r, r))
>
>     if x >= 14:
>         result = (x - r) * 3
>     else:
>         if r <= x - 6:
>             result = (x + r) // 2 + 3
>         else:
>             result = (r - x) + 5
>
>     return result if result > 4 else (r - x)
> ```
>
> Above we show an example of `x_determines_r`. This new dataset contains 600 examples in the code domain. We also carry out evaluation on this new dataset with Qwen2.5-1.5B-Instruct, its SFT and RLVR variants, and proprietary LLMs including GPT-4o and Claude-4-Sonnet. The complete results are shown below. We observe a similar trend as reported in Table 1 of the main paper, where RL still achieves superior generalization performance on the new benchmark with the presence of dependency between X and R, and exceeds the proprietary GPT-4o by a large margin.
>
> |     **Model**    |  **F1**  |
> |:----------------:|:--------:|
> | Qwen2.5-Instruct |   24.5   |
> |    Qwen2.5-SFT   |   45.6   |
> |    Qwen2.5-RL    |   58.2   |
> |  Claude-4-Sonnet | **85.1** |
> |      GPT-4o      |   44.0   |
>
>
>
> > **3. In line 101, the paper states: “infer r based on the observation $y = -1$ (abduction).” Should this instead be “infer $r$ based on the observation $x = 1, y = -1$”? Also, does abduction always consider both x and y as observations in this benchmark?**
>
> Thanks for pointing these out! The observation should indeed be $x = 1, y = -1$, and throughout this work, the abduction step defined by our benchmark should always consider the input-output pair, i.e., both x and y, as observations. We will update these points in our paper revision.

---

> > ### Author Response · Authors · 2025-11-27
> > **Followup on rebuttal**
> >
> > Dear Reviewer ZkHM,
> >
> > Thank you again for your thoughtful review of our paper. If there are any remaining concerns or points where additional clarification from our side would help your assessment, we would be happy to provide further details.

---

### Author Response · Authors · 2025-11-22
**Global Response (1/1)**

We thank all the reviewers for their informative feedback and insightful suggestions! Apart from showing additional experimental results and clarifications in separate responses, we are also actively updating them into our PDF revision for better display. We will keep the reviewers updated once the new revision is uploaded. Our main experimental updates include:

1. We have involved both human judges and an additional LLM judge to solidify the findings of our behavioral analysis on model-generated reasoning traces in Section 5;

2. We have constructed a new evaluation dataset with richer causal structure by explicitly specifying the dependency between X and R, and presented the evaluation results on this dataset.

---

> ### Author Response · Authors · 2025-11-30
> **Global Response: Updates on the PDF Revision**
>
> We thank all the reviewers and chairs for their patience. We have incorporated all the additional experiments and clarifications into the PDF revision. Below we summarize the modifications we have made, which are highlighted in blue in the PDF revision. We kindly note that the order of sections in the Appendix has also been adjusted for better display. The original Appendix Section E has been shifted to F, and we have updated the references to "Appendix E" to "Appendix F" in our previous responses accordingly.
>
> 1. **(Reviewer ZkHM)** At the beginning of the Appendix, We have added a Limitations section to reflect the discussion on our choice of evaluating support-set inference only.
>
> 2. **(Reviewer ZkHM)** In Appendix M, we have added the example function and experimental results for the new `x_determines_r` benchmark which explicitly specifies dependency between X and R.
>
> 3. **(Reviewer ZkHM)** In Section 1, we have clarified that our benchmark should always consider the input-output pair, i.e., both x and y, as observations, and corrected the typo in the original line 101.
>
> 4. **(Reviewer V1ez)** We have updated the caption of Figure 1, and expanded the description of our problem setup in Section 1. We hope this detailed formalization can help readers better understand the overall task setups.
>
> 5. **(Reviewer V1ez)** We have augmented the discussion on counterfactual example generation in both Section 2 and Appendix F.
>
> 6. **(Reviewer yTEV)** In Appendix N, we have added the experimental results for both human validation and multi-judge comparison, and summarized our findings on the high correlations between humans and LLM judges.

---

### Meta-Review · Area_Chair_KAV4 · 2026-01-10

**Summary:**

The paper proposes an executable, code-based benchmark for counterfactual reasoning that makes the abduction step explicit and verifiable. It shows current LLMs struggle with abduction, and that RL with verifiable rewards (RLVR) generalizes better than SFT to unseen functions and a natural-language counterfactual set. The rebuttal clarifies scope (support-set inference vs full counterfactual distributions), fixes notation, validates the LLM-as-judge with human correlations, and adds a new task family with explicit X→R dependency where RLVR still generalizes well.

Pros
1. Clear contribution: an explicit, testable formulation of abduction via executable tasks, with ground-truth support sets.
2. Thorough experiments across open and proprietary models; consistent finding that RLVR outperforms SFT on OOD structure.
3. Rebuttal meaningfully strengthens the paper: scope clarification, added dependency-rich tasks, and human/LLM judge alignment analyses.
4. Practical value: a reproducible framework for diagnosing where reasoning fails and for stress-testing training methods.

Cons
1. Scope is narrower than full counterfactual inference (support sets, not full posteriors); needs stronger upfront framing.
2. Mostly synthetic/code tasks; real-world grounding remains limited (the NL set is derived rather than fully naturalistic).
3. Limited theory for why RLVR generalizes; explanations are empirical/behavioral rather than formal.
4. Some presentation issues flagged by reviewers (earlier problem definition, expanded related work) need camera-ready fixes.

Although reviewer scores are mixed, the rebuttal addresses the most substantive concerns (task scope, richer causal structure, judge reliability) and reinforces the empirical claims. The benchmark fills a clear gap by operationalizing abduction with verifiable supervision, and the added analyses increase confidence in generalization findings. For camera-ready, require: explicit “support-set counterfactuals” framing in the intro, an earlier formal task definition, inclusion of the new X→R tasks and human-judge results, clarified limitations, and an expanded related-work discussion.

**Reviewer Concerns:**

1. Clarified the benchmark targets support-set inference (not full counterfactual posteriors) and will add an explicit Limitations section; notation fix and abduction uses both x and y.
2. Added a new task family with explicit X→R dependency (x_determines_r) and reported results showing RLVR still generalizes well.
3. Committed to moving a formal task definition earlier in the paper and expanding related-work (including suggested references); clarified that counterfactual examples are defined up front in the revision.
4. Pointed to Section 5 behavioral analyses contrasting SFT vs RLVR and expanded discussion (Appendix) on why RLVR generalizes; provided additional qualitative evidence.
5. Added human validation and multi-judge checks with correlation matrices (human, o4-mini, Claude-4.5-Sonnet), indicating strong alignment.
6. Articulated a rationale for executable/code setups and outlined how to increase realism (richer dynamics, noise, more complex code environments).

**Reviewer Scores:**

N/A

---

### Decision · Program_Chairs · 2026-01-26

Accept (Poster)